# Fish Oil, but Not Olive Oil, Ameliorates Depressive-Like Behavior and Gut Microbiota Dysbiosis in Rats under Chronic Mild Stress

**DOI:** 10.3390/biom9100516

**Published:** 2019-09-21

**Authors:** Te-Hsuan Tung, Yu-Tang Tung, I-Hsuan Lin, Chun-Kuang Shih, Ngan Thi Kim Nguyen, Amalina Shabrina, Shih-Yi Huang

**Affiliations:** 1School of Nutrition and Health Sciences, Taipei Medical University, Taipei 11031, Taiwan; derossi83621@gmail.com (T.-H.T.); ckshih@tmu.edu.tw (C.-K.S.); da07107003@tmu.edu.tw (N.T.K.N.); amalina.shabrina@yahoo.com (A.S.); 2Graduate Institute of Metabolism and Obesity Sciences, Taipei Medical University, Taipei 11031, Taiwan; f91625059@tmu.edu.tw; 3Research Center of Cancer Translational Medicine, Taipei Medical University, Taipei 11031, Taiwan; ycl6@tmu.edu.tw; 4Center for Reproductive Medicine & Sciences, Taipei Medical University Hospital, Taipei 11031, Taiwan

**Keywords:** chronic mild stress, depression, gut microbiota, fish oil, olive oil

## Abstract

Background: This study investigated the effects of fish oil and olive oil in improving dysbiosis and depressive-like symptoms. Methods and results: Male rats were fed normal, fish oil-rich or olive oil-rich diets for 14 weeks. Chronic mild stress (CMS) was administered from week 2. The sucrose preference test (SPT) and forced swimming test (FST) were used to determine depressive-like behavior. The SPT results revealed that the CMS, CMS with imipramine (CMS+P) treatment, and CMS with olive oil diet (CMS+O) groups exhibited significantly reduced sucrose intake from week 8, whereas the fish oil diet (CMS+F) group exhibited significantly reduced sucrose intake from week 10. The FST results showed that the immobile time of the CMS+F group was significantly less than that of the CMS-only group. Next generation sequencing (NGS) results showed CMS significantly reduced the abundance of *Lactobacillus* and increased that of *Marvinbryantia* and *Ruminiclostridium_6*. However, the CMS+F group showed an increase in the abundance of *Eisenbergiella*, *Ruminococcaceae_UCG_009*, and *Holdemania*, whereas the CMS+O group showed an increase in the abundance of *Akkermansia*. Conclusions: CMS stimuli altered the gut microbiome in depressed rats. Fish oil and olive oil exerted part of a prebiotic-like effect to ameliorate dysbiosis induced by CMS. However, only fish oil ameliorated depressive-like symptoms.

## 1. Introduction

The World Health Organization has reported that more than 350 million people worldwide have depression. Furthermore, depressive disorder is predicted to be the second leading cause of disability in 2020 [1]. Various therapies have been introduced for treating depression, however, antidepressants have severe side effects that cause low compliance among patients and even patient resistance to regular medical therapy. Consequently, discovering new therapies is extremely urgent.

Some adjunctive therapies have been discovered for the treatment of depression. Probiotic and prebiotic supplements are one potential approach. Kelly et al. [2] reported that depression can be induced in healthy rats by transplanting gut microbiota from major depressive disorder (MDD) patients. This suggests that gut microbiota might modulate brain activity and behavior. Among clinical trials relevant to depression, one randomized controlled study linked treatment with multispecies probiotics to emotional reactions of sad moods, particularly rumination and aggressive thoughts in non-depressed people [3]. Another study proved that prebiotics, which are non-digestible fibers that promote the growth or activity of beneficial microorganisms, have potential antidepressive effects [4]. The role of the microbiome–gut–brain axis in the pathology of depression has been discussed by scholars.

Associations between various types of diet and the pathology of depression have been discovered in the last two decades [5,6,7,8]. For instance, evidence has emerged that the Mediterranean-style diet has beneficial effects on neurological disorders, including stroke, depression, and cognitive impairment [9,10]. Fish oil, one of the main lipids in the Mediterranean diet, contains a high percentage of n-3 polyunsaturated fatty acid (PUFA), particularly eicosapentaenoic acid (EPA) and docosahexaenoic acid (DHA). A clinical trial that implemented a Mediterranean-style dietary intervention with fish oil supplementation demonstrated that an increase in n-3 PUFA intake reduced the severity of depression symptoms and improved quality of life [11]. The large amount of olive oil used in the Mediterranean diet has been considered to have health benefits, particularly regarding depression risk [12]. However, the association between the effects of olive oil on gut microorganisms and depressive-like behavior has not yet been elucidated in basic and clinical studies.

Scholars have concluded that the amounts and types of lipids in the diet affect the occurrence of depression [5,10,11]. C57BL/6 mice fed a high-fat diet (60% of energy (kJ) obtained from lipids, with refined palm oil as the main lipid source) for 8 weeks showed significantly decreased sociability and sucrose preference [5]. In another study, a lard-based high-fat and high-sugar diet (36% of energy (kJ) obtained from lipids) significantly reduced the frequency of social behaviors, impaired memory, and altered microbiome composition [13]. However, whether dysbiosis caused by an unhealthy saturated fatty acids-rich diet results in neurobehavioral alteration remains unclear. Therefore, we investigated whether fish oil and olive oil interventions exerted an antidepressive effect in a chronic mild stress (CMS) model and explored the potential effects of these two lipids on the intestinal dysbiosis induced by CMS.

## 2. Materials and Methods

### 2.1. Animals and Diets

In this study, male Sprague–Dawley rats (n = 43, 6 weeks old; Bio-LASCO, Taiwan) were used. The rats were housed in a temperature- and humidity-controlled room (22 °C ± 2 °C; humidity: 60%) under a 12 h light–dark cycle (light period: 08:00–20:00) and had free access to food and water. After 2 weeks of acclimation, the rats were divided into five groups (n = 8 or 9 per group): the normal control (N), CMS, CMS treated with a drug (imipramine) (CMS+P), CMS treated with a fish oil diet (CMS+F), and CMS treated with olive oil (CMS+O) groups (Figure 1). The study was conducted in accordance with institutional guidelines and approved by the Institutional Animal Care and Use Committee of Taipei Medical University (LAC-2016-0405).

The animal diets were prepared on the basis of the AIN-93M semi-purified diet composition. Three oil-based diets were used, specifically, the diets of the N, CMS, and CMS+P groups contained 4% (*w*/*w*) soybean oil, the diet of the CMS+F group contained 2% fish oil and 2% soybean oil, and the CMS+O group’s diet contained 2% olive oil and 2% soybean oil. The fish oil (Chueh Hsin Co., Taipei, Taiwan) contained 20.5% (*w*/*w*) EPA and 11.2% DHA, whereas the extra virgin olive oil (EVOO) (Laconia Greece S.A., Sparta, Greece) contained 65% oleic acid.

### 2.2. Experimental Protocols

The experiment was conducted over 14 weeks (Figure 1). Briefly, the diets containing different dietary oils were administrated during the experimental period. Except for the N group, all groups were subjected to CMS from week 2 to week 14. CMS was exerted every week by randomly applying six out of nine possible stresses. The chronic mild stresses were as follows: (1) Water and food deprivation for 12 h, (2) a 30° cage tilt for 6 h, (3) damp sawdust (250 mL water in sawdust bedding) for 24 h, (4) physical restraint for 1 h, (5) cold swimming for 1 h, (6) blank cages without sawdust for 24 h, (7) reversed rhythm circadian for 2 days, (8) living space limitation for 8 h, and (9) social stress for 12 h. The N group lived normally without being placed under any stress. Food and water were freely available. Imipramine (Sigma-Aldrich Co., Ltd., Taiwan) was administered daily to the CMS+P group through drinking water (20 mg/kg) from week 8 to week 14, as described elsewhere [14].

### 2.3. Sucrose Preference Test

The sucrose preference test (SPT) is a measure of CMS-induced anhedonia, a key depressive behavior. Briefly, the rats were fasted for 12 h and then given two bottles of water, one containing reverse osmosis water and the other containing 1% sucrose solution. Sucrose preference was calculated as the percentage of the 1% sucrose solution consumed relative to the total liquid intake.

### 2.4. Open Field Test and Forced Swimming Test

The apparatus for the open field test (OFT) consisted of a square area (50 × 50 cm^2^) with walls 40 cm high constructed from black polyvinyl chloride plastic board. The arena was lit by lights placed 145 cm above the arena and was divided into a central area (25 cm × 25 cm) and an outer area, which included the peripheral region of the arena and the wall area. During a test session, the total distance traveled and central visit duration were measured. Each test session lasted 5 min and was recorded using a video camera placed 145 cm above the arena. The videos were analyzed using ActualTrackTM software (ActualAnalytics Co., Ltd., UK).

The forced swimming test (FST) is a model of behavioral despair that is considered effective for predicting antidepressant efficacy. The study was conducted using a previously reported method with slight modification [15]. In brief, each rat was placed in a Plexiglas cylinder (37 cm in diameter and 70 cm in height) containing 50 cm of water (24 °C ± 1 °C). In the pretest session, a rat was placed in the water for 15 min to induce a state of despair and then dried with a towel and warmed in a plastic cage under a heat lamp. After 24 h, the rat was exposed to the same experimental conditions for a 5 min test session, which was recorded by HDR-SR1 (SONY, Tokyo, Japan). The videos were analyzed using software (Forced Swim Scan 2.0, CleverSys, Reston, VA, USA) to determine the time each rat spent immobile, swimming, and struggling (including climbing, escaping, and diving)).

### 2.5. Corticosterone Assay

After finishing the FST, the rats were immediately anesthetized. Blood was collected directly from the abdominal aorta, stored in prechilled ethylenediaminetetraacetic acid (EDTA)-coated blood collection tubes, and centrifuged (3000 rpm, 10 min, 4 °C). Plasma was taken and immediately stored at −80 °C until analysis. Plasma corticosterone was measured using the AssayMax™ Corticosterone ELISA kit (AssayPro, St. Charles, MO, USA), according to the manufacturer’s instructions.

### 2.6. Lipid Extraction and Fatty Acid Profile Analysis

Selected tissue samples were extracted using a modified Folch method, as previously described [16]. Twenty milligrams of prefrontal cortex (PFC) or hippocampus and 1 mL of phosphate buffered saline (PBS) were completely homogenized, and 1 mL of red blood cells was mixed with 1 mL of water. Methanol (1.5 mL) was added to 200 μL of the sample aliquot and vortexed. Subsequently, 3 mL of chloroform was added, the mixture was left for 1 h at room temperature with gentle shaking, and the liquid was then separated by adding 1.25 mL of water. The extract was incubated for 10 min at room temperature and centrifuged at 3000 rpm for 10 min. The lower (chloroform) phase was collected. Phospholipids were separated using a HybridSPE^®^-Phospholipid column (Supelco, St. Louis, MO, USA). Fatty acid methylation was performed by heating the samples at 90 °C for 1 h with boron trifluoride–methanol reagent (15%, 0.3 mL) to form fatty acid methyl ether (FAME), and the solvent was then removed using a vacuum pump. The FAME was analyzed using a TRACE™ gas chromatograph (Thermo Fisher Scientific Inc., Milan, Italy) equipped with a 30 m × 0.32 mm inner diameter (I.D.) × 0.20 μm df Rtx-2330 column (Restek, Bellefonte, PA USA) and flame ionization detector. The gas chromatograph oven temperature was initially maintained at 160 °C and then increased at 5 °C/min to 250 °C, where it was maintained for 5 min. The injector and detector were both maintained at 260 °C. Results were obtained according to the retention time of the appropriate standard GLC-455 (Supelco, St. Louis, MO, USA), and the percentage of fatty acid profiles was calculated based on the 12 different fatty acids (Appendix A).

### 2.7. DNA Extraction, Amplification, and Sequencing

Faeces were collected before the rats were sacrificed and immediately stored at −80 °C until analysis. DNA was extracted from 200 mg of faeces by using the PowerSoil^®^ DNA Isolation kit (MO BIO Laboratories, Inc., Carlsbad, CA, USA), according to the manufacturer’s instructions.

The Illumina MiSeq system and the MiSeq Reagent Kit v2 500-cycle (San Diego, CA, USA) was used to sequence the V3–V4 regions of the 16s rRNA gene extracted from rat faeces. Universal primers were removed, and low-quality reads were trimmed using cutadapt (v1.15). The paired reads were then processed using the DADA2/phyloseq workflow in the R environment. Briefly, filtering, trimming, dereplication, and denoising of the forward and reversed reads were performed using DADA2 (v1.6.0). The paired reads were then merged, and chimeras were subsequently removed. The inferred amplicon sequence variants were subjected to taxonomy assignment using the SILVA database (v132) as the reference, with a minimum bootstrap confidence of 80. Multiple sequence alignment of the amplicon sequence variants was performed using DECIPHER (v2.6.0), and a phylogenetic tree was constructed from the alignment using RAxML (v8.2.11). The frequency table, taxonomy, and phylogenetic tree information were used to create a phyloseq object, and bacterial community analyses were performed using phyloseq (v1.19.1).

### 2.8. Statistics

All data were presented as mean ± SD. Statistical analyses were performed using GraphPad Prism software, version 7.0 (San Diego, CA, USA). The data were analyzed using analysis of variance followed by Tukey’s post hoc test. The correlation between microbiota abundance and the fatty acid profile was analyzed using Pearson’s correlation coefficient; *p* < 0.05 was considered significant. Microbiota enrichment analysis was conducted using the linear discriminant analysis effect size (LEfSe) method and visualized through cladograms obtained using GraPhlAn.

## 3. Results

### 3.1. Weight Change under CMS

At the beginning of the experiment, the groups did not differ significantly in body weight [*F*(4,38) = 2.003, *p* > 0.05]. CMS stimuli were administered to the rats from week 2, and the body weight gain of the CMS-treated rats (CMS, CMS+P, CMS+F, and CMS+O groups) slowed during week 4–8 compared with that of the N group (Figure 2). A similar result was observed in week 8–14. In addition, the weight change of the CMS+P group was significantly smaller than that of the other groups during week 8–14.

### 3.2. Anxiety-Like Behavior Test

The OFT was used to identify anxiety-like symptoms in this research. We determined that the CMS rats traveled significantly smaller total distances than the N group. Fish oil and olive oil intervention restored this anxiety-like behavior, but imipramine had no effect on anxiety-like behavior in the OFT (Figure 3A). Nonetheless, no significant differences in central visit duration were discovered between all groups (Figure 3B).

### 3.3. Depressive-Like Behavior Tests

Anhedonia, a depressive-like behavior, was represented by the percentage of sucrose solution intake in the SPT. Compared with the N group, the CMS rats had a significantly lower percentage of sucrose solution intake in week 8, except for the CMS+F group. We also observed that fish oil intervention delayed the onset of depressive-like behavior in the SPT, because the CMS+F group showed significantly reduced sucrose intake at week 10. The antidepressant imipramine was administered to the CMS+P group from week 8 to week 14, and after 4 weeks (at week 12), sucrose intake trend was significantly reversed (Figure 3C). Thus, six weeks of CMS successfully induced depressive-like symptoms in this study.

As illustrated in Figure 3D, the time the rats spent immobile in the FST was analyzed and represented depressive-like symptoms. Compared with the N group, the immobile time of the CMS group significantly increased during the test (*p* = 0.001). Fish oil intervention and imipramine treatment reversed the stress-induced abnormal depressive-like behavior exhibited in the FST (*p <* 0.001).

Overactivity of the hypothalamic-pituitary-adrenal (HPA) axis was discovered in long-term corticosterone levels after the FST (Figure 3). A significant difference was revealed in the corticosterone levels between the N and CMS groups (*p* = 0.02). Fish oil, olive oil, and imipramine did not recover the abnormally high corticosterone under acute stress (Figure 3D).

### 3.4. Fatty Acid Profiles of Brain and Red Blood Cells

Different dietary lipids, such as fish oil and olive oil, can result in significantly differing fatty acid profiles in both brain regions (the PFC and hippocampus) and red blood cells. The percentage of EPA (C20:5) in the prefrontal cortex (PFC) was significantly higher in the CMS+F group than the other groups, whereas the percentage of DHA (C22:6) was unaffected by the fish oil intervention (Figure 4). We also discovered that the C18:0 in the PFC was significantly higher in the CMS+O group than the N group (Appendix A). Moreover, the EPA percentage in the hippocampus of the CMS+F group was significantly higher than that of the other groups, except for the N group. The DHA percentage in the hippocampus of the CMS+F group was significantly higher than that of the N group. We also found that the EPA percentage in the red blood cells of the CMS+F group was significantly higher than that of the other groups (Appendix A).

### 3.5. Microbiota Alteration after Different Dietary Lipid Interventions

After removing bias from variation in the sample read number, sequencing of the microbiota performed using the Miseq led to 2.2 million sequenced reads. In the N and CMS group samples, distinct clustering and separation of the CMS group from the N group were observed in the nonmetric multidimensional scaling plot within the Bray–Curtis distance methods. The results revealed clear separation of the CMS+F and CMS+O groups from the other groups (Figure 5A).

Analysis of relative abundance using the linear discriminant analysis (LDA) effect size (LEfSe) method indicated that the faecal microbiota composition differed significantly between all groups. Bacteroidaceae, Prevotellaceae, and Lactobacillaceae were significantly more abundant in the N group. *Bacteroides, Lactobacillus, Terrisporobacter, Candidatus_Stoquefichus*, and *Proteus* were significantly more abundant in the N group, whereas *Marvinbryantia, Ruminiclostridium_6, Ruminococcaceae_NK4A214*, and *Erysipelotrichaceae_ge* were significantly more abundant in the CMS group. Different dietary lipids resulted in differing microbiota compositions. At the genus level, fish oil (CMS+F group) significantly elevated the relative abundance of *Eisenbergilla, Ruminococcaceae_UCG_009*, and *Holdemania*. Nevertheless, the CMS+O group had significantly higher abundances of *Romboutsia, Akkermansia*, and *Ruminococcaceae_UCG_003* (Figure 5B,C).

### 3.6. Correlation between Fatty Acid Profiles and Microbiota

We evaluated the correlations between the relative abundance of microbiota and EPA, DHA, and arachidonic acid (AA) percentage by using Pearson coefficient correlation analysis. The results indicated that some bacterial genera were significantly correlated with the percentage of specific fatty acids in the hippocampus. *Lachnospiraceare_UCG006* was positively correlated with EPA and DHA percentages (Figure 6A,B *p* = 0.0198 and 0.04, respectively). We also determined that *Eryspelatoclostridium* was negatively correlated with EPA percentage (Figure 6C, *p* = 0.04). Notably, *Ruminiclostridium_5*, which belongs to Ruminococcaceae, negatively correlated with AA percentage (Figure 6D, *p* = 0.029) and positively correlated with EPA percentage (Figure 6E, *p* = 0.006).

## 4. Discussion

The CMS model is recommended as an effective animal model that induces depressive-like symptoms such as anhedonia, hopelessness, and despair [17,18,19]. This depression animal model simulates the progression of human depression, which is induced by continuous psychological stress. The present data demonstrated that CMS induced a depressive-like state characterized by a decrease in sucrose solution intake in the SPT and an increase in immobile time in the FST.

The CMS model also resulted in lower body weight gain in the experimental rats than the control rats. Cavigelli et al. demonstrated that four weeks of CMS reduced the body weight of rats [20]. Another study reported that six weeks of CMS significantly reduced body weight, and fluoxetine antidepressant treatment improved depressive-like behaviors without causing body weight gain [21]. Our study demonstrated that persistent psychological stress for more than 12 weeks strongly influenced body weight gain. Given that the food consumption was significantly lower after two weeks of CMS, persistent psychological stress may attenuate the appetite of rats. Notably, food consumption slowly recovered from week 10 in all CMS groups except the CMS+P group; however, the body weight of the CMS groups remained lower than that of the N group. Our data revealed that imipramine may have adverse effects such as decreased appetite [22]. In addition, the CMS intervention may have impaired the rats’ physiological metabolism [23]. Therefore, the food consumption and body weight of the CMS+P group were significantly lower than those of the other groups (Appendix A).

Various neurobehaviors, such as despair, anhedonia, loss of willingness to explore, and general locomotor activity, are considered to represent depressive-like and anxiety-like behaviors caused by various types of stress [1,21,24,25]. In this study, CMS strongly induced anxiety-like and depressive-like behaviors, including decreased total distance traveled in the OFT, percentage of sucrose solution intake in the SPT, and increased immobile time in the FST. Dietary fish oil intake increased the total distance traveled and percentage intake of sucrose solution and reduced the immobile time; however, the other main lipid source in the Mediterranean diet, olive oil, did not improve depressive-like behaviors. Notably, the total distance traveled and central visit duration of the CMS+P group did not improve at all, indicating that imipramine had a weak anxiolytic effect [26]. However, several studies reported that imipramine can improve anxiety-like behaviors in the elevated-plus-maze test [27,28]. Therefore, the anxiolytic effect of imipramine remains controversial and requires additional research.

Fish oil, which contains large amounts of n-3 PUFA, has been demonstrated to exert antidepressant effects in both preclinical and clinical studies [25,29,30,31]. Recently, a meta-analysis of 13 randomized clinical trials indicated that fish oil exhibited beneficial effects in patients with major depressive disorder [32]. In the present study, the fish oil intervention slowed the progression of depression characterized by significantly decreased sucrose intake until week 10 for the CMS+F group compared with the N group, which was two weeks later than the other CMS groups. Thus, the fish oil intervention exerted an antidepressant effect.

The most well-known pathogenesis of depression is HPA axis dysregulation [33,34]. Because of a potent endocrine mechanism, the HPA axis influences numerous other parameters related to the pathophysiology of depression, such as monoaminergic neurotransmission, synaptic plasticity, and neuropeptide activity [35]. Furthermore, HPA axis dysregulation in patients with depression was found to be associated with immune function and age-related disease [36]. In the present study, corticosterone, which is a reliable biochemical marker of HPA axis dysregulation, was detected after the FST. The results indicated that after 12 weeks of CMS, the ability of the rats to adapt to acute stress decreased significantly because the corticosterone levels of the CMS-related groups were significantly higher than those of the N group. We also discovered that the antidepressant effects of n-3 PUFA was not associated with corticosterone level, because different dietary lipids could not relieve overactivity of the HPA axis. However, we only measured the corticosterone level of the rats. Whether the various dietary lipids modulate other parts of the HPA axis, such as the functions of glucocorticoid receptors or signal transduction of the central nervous system, remains unclear.

To determine whether the intake of different lipids altered the phospholipid composition of rat cell membranes, we analyzed the fatty acid profile of phospholipids in the PFC and hippocampus. The results revealed that fish oil intake not only ameliorated depressive-like behaviors in the FST and SPT, but also changed the composition of phospholipids in the rats. This suggests that EPA may play a more crucial role than DHA in the antidepressive effect of fish oil. A meta-analysis reported that EPA exerted a stronger effect on patients with depressive disorder than those without [37]. Studies have demonstrated that more than 50% of DHA supplements do not significantly reduce the severity of depression symptoms, whereas supplements containing pure EPA or more than 50% EPA significantly improve these symptoms [30,38]. Notably, although DHA is a critical and fundamental component of the brain, only EPA was elevated significantly in the hippocampus and PFC. Conversely, DHA has various essential functions in the brain and exerted an antidepressant effect on women with postpartum depression [30]. In recent years, some scholars have mentioned that intake of free DHA and triglyceride (TG)-form DHA did not increase DHA levels in the brain or improve depression symptoms because these forms of DHA could not completely cross the blood–brain barrier and simply elevated DHA levels in adipose tissue and the heart, whereas lysophosphatidylcholine-form DHA significantly increased DHA levels in the brain [39,40].

Furthermore, our data suggest that behavior alterations may be regulated by gut microbiota. Recently, microbiota have been proven to contribute to depression through several mechanisms. For example, scholars proved that the gut microbiota directly affects the immune system through activation of the vagus nerve [41,42]. In our study, linear discriminant analysis with effect size measurement was used to determine the major bacteria in the different treatment groups. Intestinal dysbiosis in the CMS group was characterized by identifying significant taxonomical differences compared with the N group. Well-known probiotics, namely *Lactobacillus* and Lactobacillaceae, were significantly decreased in the CMS-exposed rats. The abundances of Bacteroidaceae, *Bacteroides*, and Prevotellaceae were significantly higher in the N group than in those rats with dysbiosis. *Ruminiclostridium_6* was significantly higher in the CMS group in the LEfSe analysis and was previously reported to be increased in both early-diabetic and diabetic mice [43]. The relative abundance of *Ruminiclostridium_6* was positively correlated with the levels of pro-inflammatory factors IL-17A, TNF-α, and lipopolysaccharides (LPS), but negatively correlated with the level of anti-inflammatory cytokine IL-10 in the plasma [43]. Dietary prebiotics significantly improved the inflammatory response and reduced the abundance of *Riminiclostridium_6* [43,44].

In the present study, dietary fish oil prevented depressive-like status by improving gut dysbiosis. Prebiotic chrysanthemum polysaccharide intervention was previously reported to boost the abundance of *Ruminococcaceae UCG_009* [44]. In this research, *Ruminococcaceae UCG_009* increased with the fish oil intervention in the CMS+F group. In clinical trials, high dietary fiber intake increased the abundance of *Holdemania*, which is involved in butyrate production [45]. *Holdemania* abundance also increased in the CMS+F group. Moreover, dietary olive oil changed the composition of the microbiota considerably by significantly increasing the relative abundance of *Akkermansia*, which was demonstrated to have a positive effect on obesity, insulin resistance, and diabetes [46,47]. In the present study, however, olive oil did not exhibit a positive effect on psychological abnormalities.

Pearson’s correlation analysis was performed to elucidate the associations between types of fatty acids in the hippocampus and the relative abundance of microbiota to understand the potential effects of dietary lipids on the gut microbiota. Our results indicated that *Lachnospiraceae_UCG006* abundance was positively associated with both EPA and DHA percentages in the hippocampus. We also discovered that *Ruminiclostridium_5* was significantly positively associated with EPA percentage and significantly negatively associated with AA percentage. EPA is an n-3 fatty acid and exerts an anti-inflammatory effect, whereas AA is an n-6 fatty acid and has a pro-inflammatory effect.

Although our study revealed that different dietary lipids resulted in differing gut microbiota composition, which may involve improvement of CMS-induced depression, there were still some limitations in our experiment. First, the olive oil we used only contained 65% oleic acid. Considering that oleic acid and polyphenols are two main beneficial factors of EVOO, we could have used EVOO, which contains higher levels of oleic acid. Also, more studies should be conducted to elucidate the effects of dietary lipids on the composition of microbiota, especially clinical trials. Whether dietary lipids affect the composition of the intestinal microbiota in humans still needs to be demonstrated.

## 5. Conclusions

In summary, fish oil improved psychiatric status by ameliorating gut dysbiosis in the CMS rat model. Additionally, the two main lipids in the Mediterranean diet, namely, fish oil and olive oil, were discovered to exert part of the effect exerted by prebiotics, preventing CMS-induced dysbiosis. Fish oil exerted a mild preventive effect on depression and improved the severity of depressive-like symptoms, but olive oil exhibited weaker effects than fish oil. Additional studies involving the alteration of lipid metabolites may elucidate the temporal and causal relationships between gut microbiota, depression, and dietary lipids.

## Figures and Tables

**Figure 1 biomolecules-09-00516-f001:**
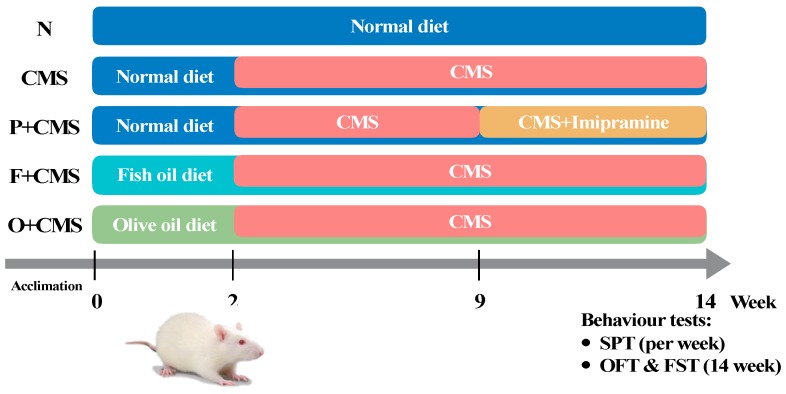
Experimental flow chart. Experimental animals were divided into five groups. CMS—chronic mild stress; SPT—sucrose preference test; OFT:—open field test; FST—forced swimming test.

**Figure 2 biomolecules-09-00516-f002:**
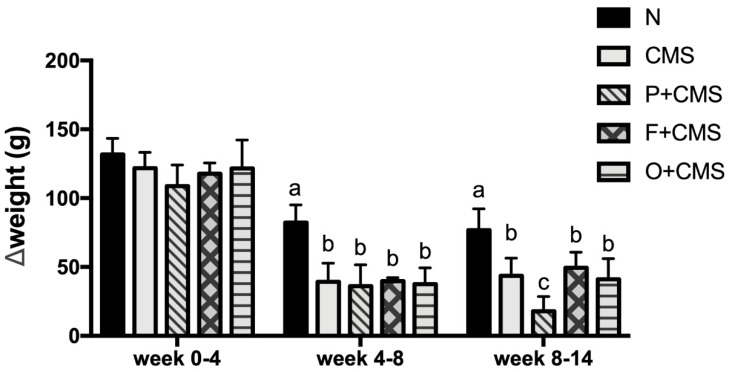
Weight change groups. The entire experimental period was divided into three parts. The group differences are displayed for each part. CMS was administered from week 2, and CMS significantly reduced the increase in weight during week 5–8. Data are expressed as mean ± SD (n = 8 or 9 per group). Values with different superscript letters are significantly different at *p* < 0.05.

**Figure 3 biomolecules-09-00516-f003:**
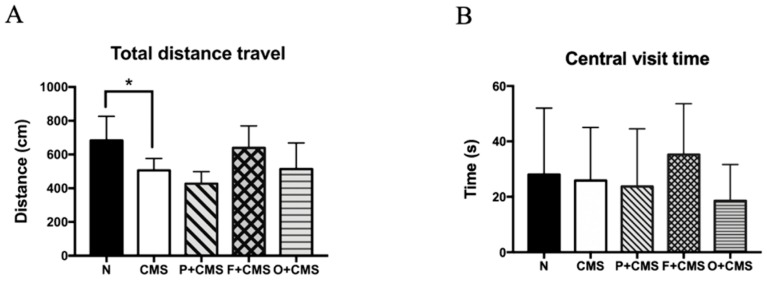
Anxiety-like and depressive-like behavioral tests. (**A**) Total distance traveled and (**B**) central visit duration in the OFT. CMS significantly reduced the total distance traveled. (**C**) Percentage of sucrose water consumed in the SPT. The CMS groups drank significantly less sucrose water. (**D**) Immobile time in the FST and (**E**) corticosterone levels in plasma. CMS significantly elevated the immobile time. Data are expressed as mean ± SD (n = 6 per group). * Significantly different using Student’s t test (*p* < 0.05). Values with different superscript letters are significantly different at *p* < 0.05.

**Figure 4 biomolecules-09-00516-f004:**
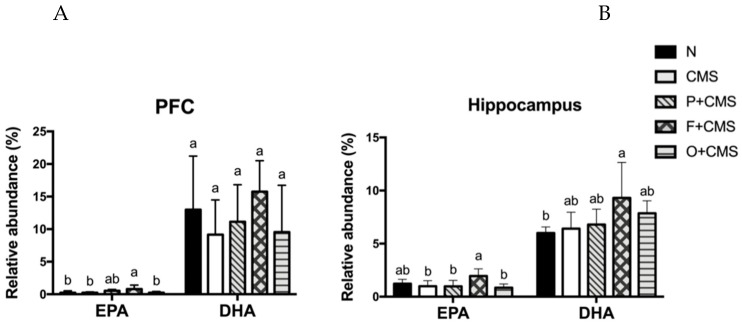
Percentages of eicosapentaenoic acid (EPA) and docosahexaenoic acid (DHA) in the (**A**) prefrontal cortex (PFC) and (**B**) hippocampus. Data are expressed as mean ± SD (n = 8 or 9 per group). Values with different superscript letters are significantly different at *p* < 0.05.

**Figure 5 biomolecules-09-00516-f005:**
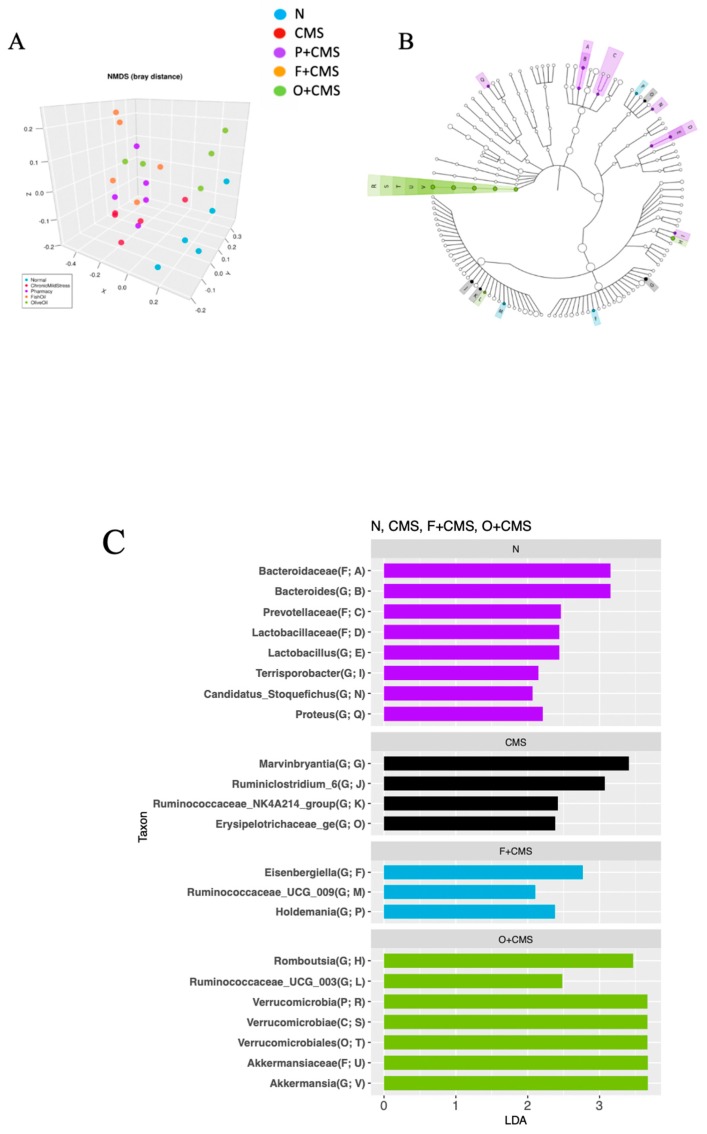
Taxonomic differences in faecal microbiota. Comparison of relative abundance across all groups. (**A**) Non-metric multidimensional scaling Bray distance methods were used to discriminate between the five groups. (**B**) LEfSe was used to identify the most differentially abundant bacteria in all groups, except the CMS+P group. The brightness of each dot is proportional to the effect size. (**C**) Only bacteria meeting a linear discriminant analysis threshold of >2 are shown (n = 5 per group).

**Figure 6 biomolecules-09-00516-f006:**
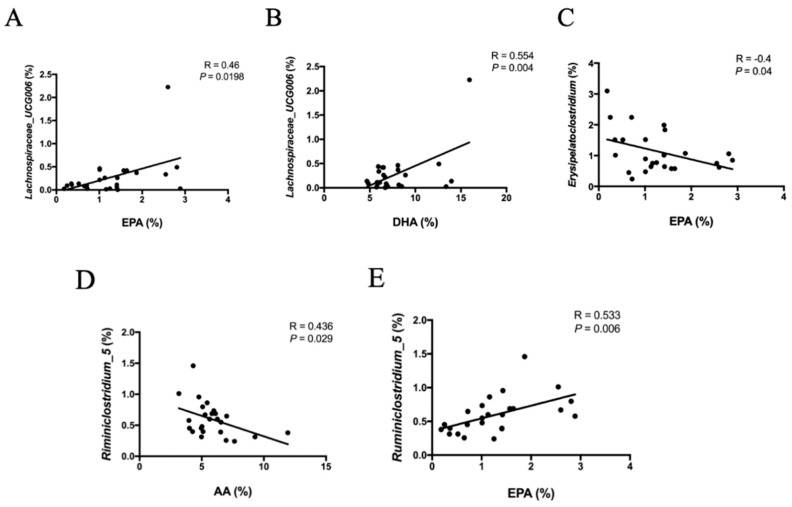
(**A**) and (**B**) Pearson correlation between relative abundance of *Lachnoapiraceae_UCG006* and percentage of EPA and DHA in the hippocampus, respectively. (**C**) Pearson correlation between relative abundance of *Erysipelatoclostridium* and percentage of EPA in the hippocampus. (**D**) and (**E**) Pearson correlation between relative abundance of *Ruminiclostridium_5* and percentage of EPA and arachidonic acid (AA) in the hippocampus, respectively. A value of *p* < 0.05 indicates a significant correlation.

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
