# Peer review of "Fish Oil, but Not Olive Oil, Ameliorates Depressive-Like Behavior and Gut Microbiota Dysbiosis in Rats under Chronic Mild Stress"

_biomolecules, 2019, doi:10.3390/biom9100516_

Round 1

Reviewer 1 Report

REVIEWED MANUSCRIPT: Fish Oil, but not Olive Oil, Ameliorates Depressive-Like Behaviour and Gut Microbiota Dysbiosis in Rats under Chronic Mild Stress”

MANUSCRIPT STATE: accepted with revision

OVERALL ASSESSMENT

The manuscript of Te-Hsuan Tung M et al. investigates the effect of two important oil (Fish Oil and Olive Oil) in depressive status in rats.

The paper its interesting and its suitable for the publication in biomolecules,but only after the authors have made a careful review.

MAJOR COMPULSORY REVISIONS

In general, I could find that one of the points to be clarified is about olive oil. In fact, the Mediterranean diet uses extra virgin olive oil with a high concentration of oleic (more than 70%) and polyphenols.

MINOR ESSENTIAL REVISION

Line 63-66: Review space, pagination and revise the sentence. Line 67-69: Revise the sentence in English (found to significantly reduce…ecc.) Line 85-86 insert CMS+P; CMS+F and CMS+O instead P+CMS; F+CMS and O+CMS ad do thesame on the all manuscript. Line 86: delete that of the

Author Response

Response to reviewers' comments:

Reviewer #1

Q1. The Mediterranean diet uses extra virgin olive oil with a high concentration of oleic (more than 70%) and polyphenols.

Answer: Thanks for reviewer suggestion. We used the fresh extra virgin olive oil (EVOO) (Laconia Greece S.A., Sparta, Greece) which conform to the EVOO criteria of International Olive Council (IOC) including acidity < 0.8%, peroxide value < 20 mEq, K270 value < 0.22, K232 value < 2.50 and ΔK < 0.01. We also confirmed the criteria of EVOO, and it doesn’t include the percentage of oleic acid. The percentage of oleic acid of olive oil were between 55.0-83.0 based on the different area and olive species. In this experiment, the EVOO contained 65% oleic acid and it confirmed the criteria of olive oil. Given that oleic acid is one of the important factors of olive oil, we will consider using other kinds of EVOO which contained higher percentage (>70%) of oleic acid to do the future study. In addition, we have a mistake in the original manuscript. We presented that the source of fresh EVOO is the importer, not the manufacturer and we have modified it in the revised manuscript (Page 2, Line 83-84).

Reference

International Olive Council designations and definitions of olive oils. http://www.internationaloliveoil.org/estaticos/view/83-designations-and-definitions-of-olive-oils

International Olive Council (IOC) and California trade standards for olive oil. http://cesonoma.ucanr.edu/files/27262.pdf

Q2. Line 63-66: Review space, pagination and revise the sentence.

Answer: It has been modified in our revised manuscript. (Page 2 Line 59-62)

Q3. Line 67-69: Revise the sentence in English (found to significantly reduce…ecc.)

Answer: It has been modified in our revised manuscript. (Page 2 Line 62-64)

Q4. Line 85-86 insert CMS+P; CMS+F and CMS+O instead P+CMS; F+CMS and O+CMS ad do the same on the all manuscript.

Answer: It has been modified in our revised manuscript. (Page 2 Line 80-81)

Q5. Line 86: delete “that of the”

Answer: It has been modified in our revised manuscript. (Page 2 Line 81)

Reviewer 2 Report

Tung et al report on a 14 week intervention trial in a total of 43 male rats, comparing the effects of a control diet (two groups), imipramine, a fish oil-rich diet, and an olive oil-rich diet on depressive-like behavior and gut microbiota dysbiosis in groups of 8 – 9 rats each, i.e. a total of 5 groups of rats. After two weeks, four groups of rats were additionally exposed to chronic mild stress (CMS) by various means, while one control group remained unstressed. Diets contained either 4% soybean oil, or, for the fish oil group, 2% fish oil and 2% soybean oil, or, for the olive oil group 2% olive oil and 2% soybean oil. Rats had free access to their respective diets. Anxiety was measured with the open field test, depressive-like behavior was assessed with the sucrose-preference test and the forced-swimming test, while gut microbiota were assessed by DNA sequencing. Tissue and red blood cell fatty acid composition was assessed using a method established in the authors’ lab. Rats exposed to fish oil had less inactive time in the forced-swimming test than the control group. Chronic mild stress altered the gut microbiota in the control group differently than in the fish oil group, and again differently in the olive oil group. The changes in gut microbiota were correlated with fatty acid composition, and many were found – which, in the absence of a hypothesis, must be considered exploratory findings. The authors concluded: “CMS stimuli altered the gut microbiome in depressed rats. Fish oil and olive oil exerted prebiotic-like effect to ameliorate dysbiosis induced by CMS. However, only fish oil ameliorated depressive-like symptoms.”

Introduction is lengthy and could be shortened for the parts relating general knowledge. It properly develops the research question, however. Methods are clearly and understandably reported. Results are reported transparently and at length. Discussion is lengthy again, but misses a limitations para. In light of the questionable fatty acid measurements in brain (see below), Discussion should be entirely rewritten, and shortened for the parts on the questionable data, and the pertinent limitations should be highlighted.  

Major Point: Trials in animals rarely come with a primary endpoint and a proper case estimate. This means that it is impossible to tell, whether some effects were not found because they were too small to be found, or if the study was too small to find them. This serious limitation should at least be discussed.

The fatty acid changes in red cells and brain are difficult to comprehend. After olive oil, 18:1 did not increase, although it is known, at least in humans, that red blood cells 18:1 reflects dietary intake to some degree (PMID 27632919). Likewise, after fish oil, EPA, but not DHA, increased in red blood cells and in PFC (prefrontal cortex?), while both increased in hippocampus. Brain is thought not to contain much, if any, EPA. So, one very simple explanation of the findings would be that the authors did not properly clean the brain from blood. Taken together, the fatty acid changes are very hard to explain or even believe. This makes the extensive correlating of changes in gut microbiota to fatty acid changes even more questionable.

Minor points: All abbreviations should be explained, e.g. in a glossary. Otherwise, the ms is very hard to follow.

Author Response

Response to reviewers' comments:

Reviewer #2

Q1. Introduction is lengthy and could be shortened for the parts relating general knowledge.

Answer: Introduction has been shortened in our revised manuscript. (Page 1-2)

Q2. Discussion is lengthy again, but misses a limitations para. In light of the questionable fatty acid measurements in brain (see below), Discussion should be entirely rewritten, and shortened for the parts on the questionable data, and the pertinent limitations should be highlighted.

Answer: Discussion has been entirely rewritten and shortened for the parts on the questionable data (Discussion, Page 9-11), as well as a limitations para has been added (Page 11 Line 369-374) in our revised manuscript.

Q3. Trials in animals rarely come with a primary endpoint and a proper case estimate. This means that it is impossible to tell, whether some effects were not found because they were too small to be found, or if the study was too small to find them. This serious limitation should at least be discussed.

Answer: Thanks for reviewer suggestion. In this experiment, 8-9 rats were used for each group. If we want to have effective statistical results, the minimum numbers of rats shouldn’t be lower than 5 per group. And if we use over 10 rats per group, it will violate the 3R principle (replacement, reduction and refinement). In this situation, 8-10 per group will be better to choose. According to the Sugasini et al. (2017), they used 8 mice per group. Robertson et al. (2016) also used 10 rats per group. Therefore, we believed our study as an animal study, we already had enough number of rats to make a statistical meaning. In spite of we focused on the animal model, it would be better if the clinical trials will be conducted. We have added a paragraph in the limitation part of our revised manuscript. (Page 11 Line 372-374)

Reference

Sugasini D, Thomas R, Yalagala PCR, Tai LM and Subbaish PV: Dietary docosahexaenoic acid (DHA) as lysophosphatidylcholine, but not as free acid, enriches brain DHA and improves memory in adult mice. Scientific Reports 2017, 7(1):11263.

Robertson RC, Oriach CS, Murphy K, Moloney GM, Cryan JF, Dinan TG, Ross RP and Stanton CS: Omega-3 polyunsaturated fatty acids critically regulate behaviour and gut microbiota development in adolescence and adulthood. Brain, Behavior, and Immunuty 2016, 59:21-37

Q4. The fatty acid changes in red cells and brain are difficult to comprehend. After olive oil, 18:1 did not increase, although it is known, at least in humans, that red blood cells 18:1 reflects dietary intake to some degree (PMID 27632919).

Answer: Thanks for reviewer suggestion. We checked the papers about olive oil intervention, although a lot of papers reflect the dietary intake of olive oil in the fatty acid profiles of erythrocyte. There are still some papers showed no significant change in red blood cells. According to Kontogianni et al. (2013), 6-weeks intervention of extra virgin olive oil (81% oleic acid) didn’t change the fatty acid profiles of human erythrocyte. However, in our experiment, the percentage of oleic acid (C18:1) in CMS+O group has an increasing trend compared to other stressed groups, but there was no significant difference between CMS+O group and other stressed groups (Supplementary table 2).

Reference

Kontogianni MD, Vlassopoulos A, Gatzieva A, Farmaki AE, Katsiougiannis S, Panagiotakos DB and Kalogeropoulos N: Flaxseed oil does not affect inflammatory markers and lipid profile compared to olive oil, in young, healthy, normal weight adults. Metabolism 2013, 62:686-93.

Q5. Brain is thought not to contain much, if any, EPA. So, one very simple explanation of the findings would be that the authors did not properly clean the brain from blood.

Answer: Thanks for reviewer suggestion. We have totally checked all the results again. After re-analyzing and calibrating, we found there were some overlapping area with the peak represented EPA. And we separate the overlapping area manually and re-integrate. The results have been modified in our revised manuscript. (Page 7, Figure 4; Page 18, Supplementary table 1)

Q6. This makes the extensive correlating of changes in gut microbiota to fatty acid changes even more questionable.

Answer: The correlations are conducted between the relative abundance of microbiota and fatty acid profiles in hippocampus. We are not presenting the correlation between fatty acids in prefrontal cortex and microbiota.

Q5. All abbreviations should be explained, e.g. in a glossary. Otherwise, the ms is very hard to follow.

Answer: Thanks for reviewer suggestion. They have been modified in our revised manuscript. (Page 7, Line 227)

Round 2

Reviewer 2 Report

My comments have been dealt with, I have no further comments.